# Outdoor environmental comfort evaluation for retail planning in a tropical business district using Integrated Environmental Modeller

Po-Yen Lai[1]*, Wee Shing Koh[1,2], Harish Gopalan[1], Huizhe Liu[1], Dias Leong[3], Hyosoo Lee[3], Johnathan Goh[3], Jiun Yeu Lim[1], Jacob Ang[3], Gibert Peh[3], Gilbert Cher[3], Cheng Hui Eng[3], Jia Li Goh[3], Edmund Tan[3], James Tan[3]

1 Institute of High Performance Computing (IHPC), Agency for Science Technology and Research (A*STAR), Singapore, Republic of Singapore, 2 Department of Physics, National University of Singapore, Singapore, Republic of Singapore, 3 Smart District/New Estates 2/Urban Design & Architecture Divisions, JTC Corporation, Singapore, Republic of Singapore

* lai_po-yen@ihpc.a-star.edu.sg

**Data Availability Statement:** All simulation conditions and relevant data are within the paper. The CAD models used in simulations in the paper

## Abstract

This research proposes a simulation-based assessment of outdoor thermal and acoustic comfort for a planned business urban district in Singapore for retail planning using a customized OpenFOAM-centric multi-physics environmental simulation platform called the Integrated Environmental Modeller (IEM). IEM was employed to simulate the coupled impacts of solar radiation on wind and air temperature and wind and air temperature effects on traffic noise propagation in the district on the equinox and solstice day of the hottest period. Using IEM simulation results, we computed the thermal and acoustic comfort acceptability indicators derived from local field studies' results. The spatial distribution of environmental comfort acceptability indicators in the worst-case scenario can be used to distinguish the zones exposed to thermal or noise influence. The noise-affected zones are near the main roads and overlap a part of the thermal-affected area. The thermal-affected area is almost everywhere in the studied sites in the worst-case scenario. Having outdoor retail spaces with both poor thermal and acoustic comfort is not recommended if the thermal and acoustic comfort cannot be improved simultaneously. For the high-level retail planning, a simplified parametric analysis considering solar irradiance blockage and wind speed enhancements, is provided. Considering the worst-case scenario, $\geq$50% thermal acceptability can be achieved by blocking 54%-68% solar irradiance among the pedestrian thoroughfares and the retail spaces. Coupled together, blocking the solar irradiance and enhancing the wind speed can further improve thermal comfort locally. These results can guide the retail mix (e.g., al fresco restaurants, pop-up kiosks etc.) near high footfall areas and provide reference for future plans combining landscape and infrastructure, (e.g., trees with shelter walkways, green walls with outdoor ventilation fans etc.) taking into account the environmental acceptability of people working in or visiting the tropical urban district.

are available to download in the Supporting Information files.

**Funding:** This research is supported by A*STAR under its URBAN & GREENTECH OFFICE (UGTO) CONSULTANCY FUND (Grant No.: UGTO/CF/01). This development work was undertaken as a part of the project "Modeling of Air Flow, Thermal and Chemical Gas Dispersion Towards Next Generation Port (Tuas Maritime Hub)", funded by the Singapore Maritime Institute (SMI-2016-MA-04). The conclusions put forward reflect the views of the author alone, and not necessarily those of the funding institution. The funders had no role in study design, data collection and analysis, decision to publish, or preparation of the manuscript.

**Competing interests:** The authors have declared that no competing interests exist.

## 1. Introduction

Over the past two decades, rapid urbanization in Singapore has affected human livability in outdoor thermal and acoustic environments [1]. Urban corridors and high-rise urban structures absorb the intense solar radiation from the sun in the day and radiate heat at night. This induces a higher urban air temperature than that in rural areas, producing thermal stress [2], i.e., urban heat island (UHI) [3, 4] which can be particularly observed in tropical environments [5]. As modern architectures and population exponentially grows, urban transportation needs have also vastly increased. With high road density and heavy traffic, traffic noise accounts for up to 80% of whole noise sources in urban areas [6]. Thus, the urban area is significantly noisier than the rural area, i.e., urban noise island (UNI) [7, 8]. UHI and UNI coexist and seriously affect urban areas' societal life and health conditions [7]. Therefore, assessing thermal and acoustic environments for improving the urban livability of planned districts in Singapore is essential in urban planning and architectural design.

Microclimate variables (solar radiation, air temperature, and wind speed) and noise propagation together characterize the thermal and acoustic environments in urban areas [9]. To conduct thermal and acoustic environmental studies and assessments, numerical simulation has been widely used [10]. Such simulations can provide detailed information of the physical variables of interest at any point within the domain at different scales [11, 12]. Considering the typical weather condition in Singapore: a sunny day in the tropics with low average wind speeds, buoyancy-driven turbulence induced by solar radiation is the most critical parameter that influences urban microclimate simulations [13–16] in which the core physical model is the wind-thermal interaction between wind dynamics and heat transfer on building and ground surfaces in the urban environment. Meanwhile, wind and thermal shear also impact the traffic noise levels in dense canopies [17, 18]. Therefore, a coupled wind-thermal and noise propagation model is valuable to simulate and evaluate the thermal and acoustic environment in dense urban spaces for lower wind speed conditions [19]. While, by using existing simulators (e.g., Rayman [20] SOlar Long Wave Environmental Irradiance Geometry model (SOLWEIG) [21], and ENVI-met [22]), many environmental studies have been done on simulations of solar radiation, wind dynamics, and noise propagation separately or solar-wind at most, there were no reported studies on the coupled solar-wind-noise simulations to our best knowledge. Hence, a novel coupled solar-wind-noise modeling tool, i.e., Integrated Environmental Modeller (IEM), has been developed to conduct coupled solar-wind-noise simulations in Singapore at different spatial scales from individual buildings to districts and regions. The IEM framework integrates environmental models, including solar radiation [13], wind dynamics [23, 24], and noise propagation [25], to captures the correlation of several physical schemes in simulations. The outputs of such microscale simulations enable further evaluation of thermal and acoustic comfort in the Singapore's urban environment.

To adequately evaluate thermal comfort in outdoor spaces, the human physiological thermoregulation mechanisms, subjective psychological perceptions, cultural backgrounds, and weather conditions should be taken into account [26, 27]. Several thermal comfort indices have been proposed in different scopes of application, including Physiological Equivalent Temperature (PET) [28], Standard Effective Temperature (SET) [29], Universal Thermal Climate Index (UTCI) [30], Predicted Mean Vote (PMV) [31], Thermal Sensation Vote (TSV) [32], to name a few. To realize a fair thermal comfort assessment in a tropical climate, it is better to consider the results from local field studies in the existing thermal comfort index, e.g., [33, 34]. Based on the statistical results from the field studies conducted in the Singapore's outdoor environment [34], such work proposed a correlation between TSV and operative temperature derived as a linear regression formula. TSV depends on the local microclimate factors

and human physiological thermoregulation on the Singaporean background. The corresponding thermal acceptability was further derived as a function of TSV using binary logistic regression to quantify the occupants' reaction to the thermal environment.

Researchers commonly recognized the noise level as the acoustic comfort index, e.g., World Health Organization (WHO) noise guideline [35]. Similar to thermal acceptability, the research team proposed a relationship between the noise acceptability and noise level to quantify the occupants' reaction to the environmental noise according to the field study conducted in construction site offices in Hong Kong [36]. In a modern urban area, the main component of environmental noise comes from road traffic [6] Traffic noise modeling is essential to evaluate the acoustic comfort in urban spaces by calculating the noise mapping, which helps qualitatively identify the noise level at a receiver position due to traffic emission source [37]. Traffic noise modeling considers traffic volume (e.g., average vehicle speed), composition (e.g., proportion of large vehicles), road geometry, and absorption features of ground cover in the calculation. The Calculation of Road Traffic Noise (CRTN) [38] is the first systematic scheme developed to predict the noise level due to road traffic. The CRTN method has been applied to predict the traffic noise and validated experimentally in Singapore [25].

This research is the first to propose a simulation-based workflow for simultaneously assessing the thermal and acoustic environmental comforts of a planned urban district in Singapore. We conducted solar-wind-noise simulations based on a three dimensional (3D) urban district modeled by architects on equinox and solstice day during the hottest period of Singapore weather. Material properties, terrains, building appearance with 3D geometric details, and the worst-case peak period road traffic noise sources were considered in simulations. Using the local survey-based sensation models [34, 36], we further evaluate the acceptability of thermal and acoustic comfort of pedestrian thoroughfares and the retail spaces based on outputs from IEM. Moreover, we perform the parametric study of the decreasing solar irradiance and the increasing wind speed to show room for improvement of outdoor environmental thermal comfort theoretically. The evaluation provides the guidance of high pedestrian footfall areas for retail and event planning (i.e., al fresco dining, popular pop-up retail kiosks with long queues, outdoor theme business community events, etc.) that considers thermal and acoustic comfort in the new business district. This workflow provides a reference framework for incorporating environmental comfort modeling into future landscape-infrastructure planning as part of early stage architectural design. In this paper, the methodologies, including numerical approaches and the corresponding theoretical basis, are described in Sec. 2. The numerical results, the analysis, and the discussion are presented in Sec. 3. Finally, the concluding remarks on the work are given in Sec. 4.

## 2. Methodology

### 2.1 Main features of study areas and environmental conditions

**2.1.1 Main features of the business district.**   The study area, as shown in Fig 1(A) is a coastal district in the north-eastern region of Singapore. The terrain is relatively flat, with an elevation of 15.6 m above sea level. A waterway crosses through the south part of the district. The district's planned business park is marked in white, and the various locations are referred to by their respective lot numbers. Fig 1(A) shows the computer-aided design (CAD) view of the urban district where the business park is in the region marked within the black box. Fig 1 (B) represents the zoom-in feature of the marked region. For outdoor comfort evaluations, four study sites with high pedestrian or human traffic are marked in Fig 1(B), which are respectively (i) an east-west boulevard that connects the business park buildings to the waterfront ("campus boulevard"), (ii) a north-south trail that was converted from an old road to a

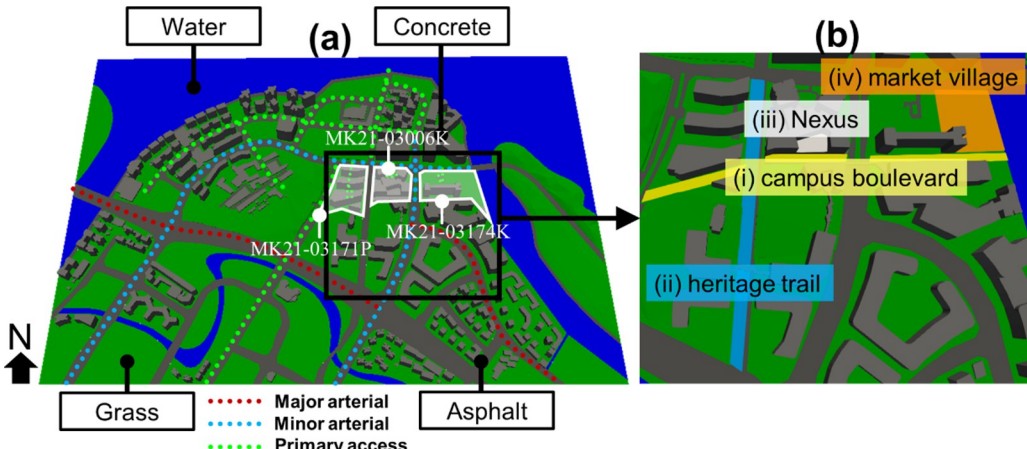

**Fig 1.** **(a)** The schematic illustration of the studied urban district in the north-east region of Singapore, including the four primary materials assigned to the 3D geometry and 3 different road categories. White portions are the planned business park with the specific lot numbers MK21-03171P, MK21-03006K and MK21-03174K. **(b)** The zoom-in feature of the study sites with high pedestrian traffic for the evaluation of comfort factors: (i) campus boulevard, (ii) heritage trail, (iii) Nexus, and (iv) market village.

pedestrian-only walkway ("heritage trail"), (iii) an open podium under a mid-rise office building ("Nexus"), and (iv) an outdoor waterfront plaza to hold gatherings and events for tenants and their customers or families ("market village"). The sites proposed in this study are either the main pedestrian thoroughfares or gathering areas for outdoor community building and events. Taking advantage of the potential high pedestrian footfall, outdoor retail options are planned along or within these sites. For example, al fresco restaurants and pop-up retail kiosks may be set up around these sites, where visitors spend their time on various shopping, leisure, and dining activities. Thermal and acoustic comfort assessments in this work can be useful for retail planning to ensure that the outdoor areas are comfortable and attractive for outdoor dining, waiting for takeaways, queuing and shopping.

**2.1.2 Environmental conditions for simulations.** In Fig 1(A), the district's urban surfaces are grouped by materials, i.e., concrete, asphalt, grass, and water, which represents the appropriate boundary conditions for IEM coupled simulation. In the various simulations, buildings, roads, greenery, and water body were assigned concrete, asphalt, grass, water in which the surface of each object has been defined own optical, thermal, acoustic properties. Table 1 gives the material properties used in this study.

The district's main roads classification is schematically represented by colored dashed lines in Fig 1(A). In Singapore, main roads are classified into five categories: expressway, major arterial, minor arterial, primary access, and local access from high to low traffic flow. This classification is purposed for the developer to establish the buffer requirements of surrounding buildings from the road [39] and used to define the traffic noise source in simulations. Table 2

**Table 1. Material properties used in this paper.**

| Material | albedo | emissivity | heat absorptivity | sound absorptivity |
|---|---|---|---|---|
| asphalt | 0.16 | 0.93 | 0.93 | 0.1 |
| concrete | 0.21 | 0.85 | 0.79 | 0.1 |
| grass | 0.39 | 0.98 | 0.61 | 0.1 |
| water | 0.14 | 0.96 | 0.86 | 0.1 |

**Table 2. Road categories used in this paper.**

| Road Category | Hourly-averaged equivalent noise level (dB) |
|---|---|
| Major Arterial | 71 |
| Minor Arterial | 70.2 |
| Primary Access | 68.3 |

shows that the categories used in this study are defined according to the numerical setup [25] in which the noise source is based on the hourly-averaged equivalent noise level during peak hours derived from measurements. Major arterials (i.e., 71 dB), shown in red, predominantly carry through traffic from one region to another, forming primary avenues of communication for urban traffic movements. Minor arterials (i.e., 70.2 dB), shown in blue, distribute traffic within the major residential and industrial areas. Primary accesses (i.e., 68.3 dB), shown in green, constitute the link between local accesses and arterial roads. In this study, the geometry description of the noise source was modeled as a finite line source at 1 meter above the center of the road surface.

The climate in Singapore is hot and humid, with a low average wind speed and abundant solar irradiation throughout the year [40]. The typical meteorological year (TMY) data [41] for Singapore was adopted as the data source in this study. Two representative days in the hottest period (i.e., from March to June in which the air temperature is commonly greater than the mean daily maximum temperature of 32°C based on the climatological reference period between 1991 to 2020 [40]) in Singapore were selected for this study, which are equinox and solstice days respectively. The equinox day was a sunny day on 20 March, where the direct normal irradiance (DNI) mostly dominates over the diffused horizontal irradiance (DHI). Fig 2 (A) shows the selected hourly-averaged climate data of sky longwave irradiance, direct normal irradiance, and diffuse horizontal irradiance on the equinox day, respectively. The solstice day was a cloudy day on 21 June. Fig 2(B) shows the corresponding meteorological data in which the radiation was diffusely dominated.

Considering that winds in Singapore are generally light, the daily average wind speed and direction were adopted in this study for simplicity. To make sure that the assessments in this study are certificated by the local policy, we adopted the Building and Construction Authority's (BCA) Green Mark scheme [42] as the wind conditions (including averaged wind speed and direction, reference height, and aerodynamic roughness length), which is the compulsory

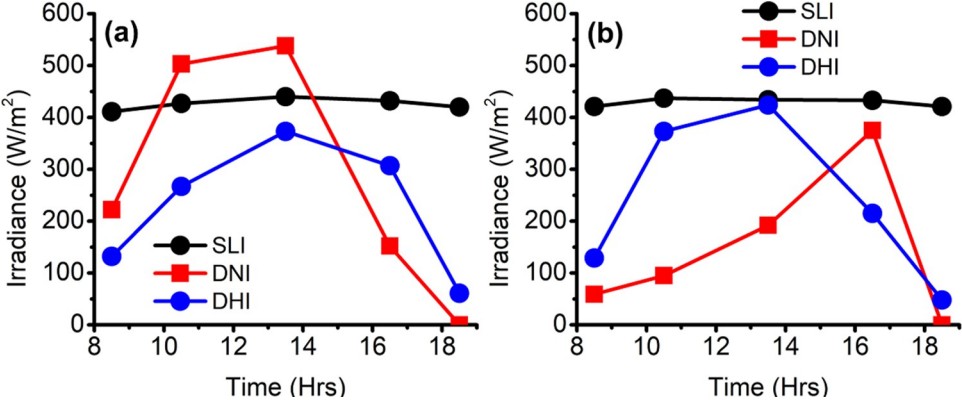

**Fig 2.** The selected hourly-averaged metrological data [41] of sky longwave irradiance (SLI), direct normal irradiance (DNI), and diffused horizontal irradiance (DHI) on **(a)** the equinox (sunny day) and **(b)** the solstice (cloudy day).

**Table 3. Climate conditions of wind applied to this paper.** The expected values follow the BCA Green Mark [42] at the reference height of 15 m based on the National Environment Agency's 18-year data.

| Date | mean speed | mean direction |
|---|---|---|
| | (m/s) | |
| Equinox (sunny day) | 2.9 | South |
| Solstice (cloudy day) | 2.8 | North-East |

requirement for green building in Singapore. Table 3 shows the average daily wind speed and direction in selected days according to the BCA Green Mark scheme [42] to evaluate green buildings in Singapore, where the values are expected at the reference height of 15 m.

## 2.2 Integrated Environmental Modeller

The Integrated Environmental Modeller (IEM) [13, 23–25] has been developed as an integrated multi-physics urban microclimatic modeling tool for urban environment evaluation customized for the tropical climate. IEM can simulate the coupling between solar irradiance, wind flow, air temperature, and traffic noise propagation. These physical models in IEM have been validated for solar radiation [13], wind dynamics [23, 24], and traffic-noise simulations [25] under the climate conditions in Singapore. Fig 3 shows the workflow of IEM. Two different methods are available for calculating solar irradiance: ray-tracing and discrete ordinates models. The ray-tracing approach was used for standalone solar irradiance studies, while the discrete ordinate models were used for wind-thermal computations to keep the computational cost low. Furthermore, the Perez all-weather sky model [13, 43] was incorporated as it has the best overall performance over a wide range of locations [44] and is one of the most suitable transposition models to predict solar irradiance for Singapore [45, 46]. Using the solar heat solver in IEM, the solar irradiance values on all surfaces in the studied system can be generated and passed to the aerodynamic simulation as input parameters.

The aerodynamic model in IEM was developed within OpenFOAM®, a finite-volume computational fluid dynamics (CFD) platform. Buoyancy effects were modeled using the

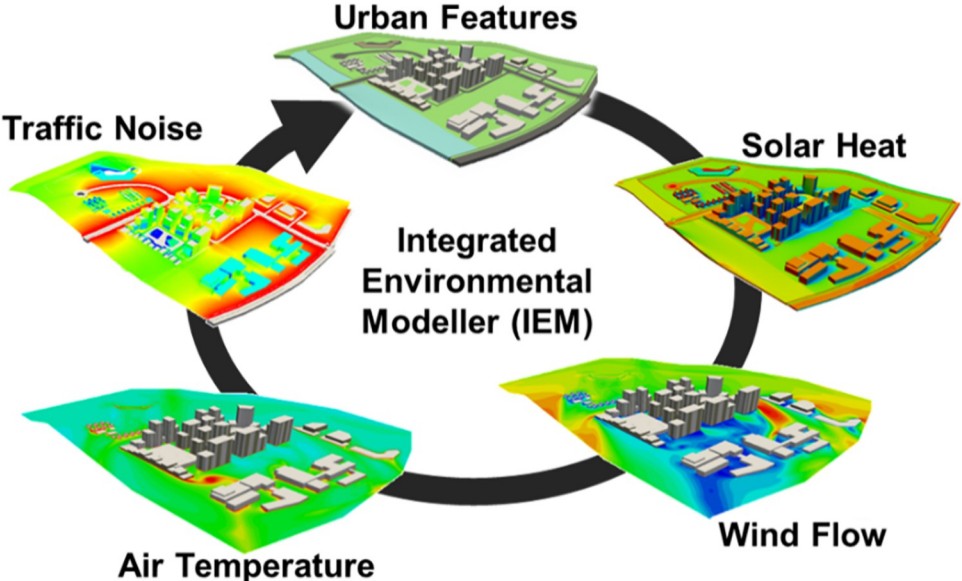

**Fig 3. The workflow of coupled simulations using IEM.**

Boussinesq approximation. The steady-state Reynolds-averaged Navier-Stokes (RANS) method was used for turbulence modeling to reduce the computational cost [24]. The turbulence effect was modeled using Wray-Agarwal (WA) one-equation model [24, 47, 48]. The turbulent external flows around the buildings of concern were resolved by solving the RANS equation. The Monin-Obukhov similarity theory (MOST) [49] was applied to specify the boundary conditions for wind, temperature, and turbulent viscosity [50]. All building surfaces and ground were set as non-slip walls. Atmospheric boundary layer profile for neutral flow was used at the inlet of the computational domain. Surface-Energy-balance (SEB) model determines the heat transfer between the wind flow and the building surface. Solar shortwave radiation, thermal longwave radiation, and convective and convection heat transfer are the main mechanisms of the SEB model. The finite volume Discrete Ordinates Method (fvDOM) [51] was adopted to simulate the longwave radiation exchange between the urban surfaces and the sky. Using the aerodynamic model in IEM, the wind speed and air temperature values in the climate space can be generated and passed to noise simulation as input parameters.

The noise propagation model in IEM was developed based on the Calculation of Road Traffic Noise (CRTN) [52] coupled with the atmospheric refraction model [53] for accessing meteorological effects. Once the turbulent wind flow has been simulated using the aerodynamic model in IEM, the calculated wind speed, lapse rate, and wind shear will be passed to the atmospheric refraction model. The atmospheric refraction effects on noise propagation are calculated and considered in the CRTN model. The noise level of the area of concern due to distance attenuation, ground absorption, screening, and site layout effects can be evaluated using the CRTN model. This approach [25] allows the adoption of a set of unstructured surface mesh to represent arbitrary 3D building geometry instead of just an extrusion of a building footprint.

## 2.3 Environmental comfort indices

Thermal indices have various degrees of dependence on environmental variables, including air temperature, wind speed, relative humidity, thermal radiation heat, and personal variables, including clothing, metabolic heat, mentality, and state of health. Because the empirical fits of these variables to derive thermal indices (e.g., predicted mean vote) are complicated, we proposed to use a compact thermal index, operative temperature ($T_O$). $T_O$ can be derived from the direct measurement of air temperature using a thermometer, mean radiant temperature using a globe thermometer, and wind speed using an anemometer. According to the standard from ISO 7726:1998 [54], the mathematical form of $T_O$ (˚C) is written as

$$T_O = \frac{T_{air}\sqrt{10U} + T_{MRT}}{\sqrt{10U} + 1},$$ (1)

where $T_{air}$ (˚C) is air temperature, $U$ (m/s) is wind speed, and $T_{MRT}$ (˚C) is mean radiant temperature. By definition [55], $T_{MRT}$ (˚C) is the effective temperature and relates to the radiation received from all the incoming angles and can be written as

$$T_{MRT} = \sqrt[4]{\sum_k \frac{F_k}{\sigma}\left[(1-\alpha_p)Q_{Sk} + \varepsilon_p Q_{Lk}\right]} - 273.15.$$ (2)

where $\sigma$ is Stefan-Boltzmann constant, $\alpha_p = 0.3$ is the effective albedo value for a person, and $\varepsilon_p = 0.9$ is the effective emissivity for a person. $Q_{sk}$ ($W/m^2$) is the received shortwave radiation, $Q_{LK}$ ($W/m^2$) is the received longwave radiation, and $F_k$ is angle factor form in $k$ direction. In this study, the evaluation of $T_{MRT}$ is on the pedestrian level (i.e., 1.5 m apart from the ground).

To reveal the relationships between environment/personal variables and the human thermal comfort sensation, ASHRAE [32] defined the unitless index as the 7-point scale (−3 for

cold, −2 for cool, −1 for slightly cool, 0 for neutrality, 1 for slightly warm, 2 for warm, and 3 for hot) called thermal sensation vote (TSV). Considering the average metabolic and clothing conditions in Singapore, a linear relationship [34] between TSV and $T_O$ was derived as

$$TSV = 0.356T_O - 10.21. \tag{3}$$

According to the on-site survey in Singapore [34], the relationship between the percentage of thermal acceptability (*PTA*) and *TSV* was achieved using binary logistical regression, i.e.,

$$PTA = \frac{\exp(-1.118TSV + 2.853)}{1 + \exp(-1.118TSV + 2.853)}. \tag{4}$$

In this study, Eq 4 was adopted to represent human acceptability with outdoor thermal conditions directly.

Noise intensity is another factor affecting human acceptability with outdoor environmental conditions. Unlike thermal comfort, the noise level is usually the only metric to define human acoustic comfort. According to the published literature [36], the logistic regression model of noise acceptance has been derived from on-site surveys and measurements in Hong Kong. Similar to Eq 4, the relationship between the percentage of noise acceptability (*PNA*) and noise level (*L*) (dB) can be written as

$$PNA = \frac{\exp(-0.55L + 34.5)}{1 + \exp(-0.55L + 34.5)}. \tag{5}$$

Eq 5 is derived according to the equivalent continuous noise level sourced from the construction site and measured in the construction office, where the acceptable noise level is 65 dB [36]. According to the guideline from Sustainable Mobility Initiative for Local Environment (SMILE) [5], traffic noise can be regarded as the primary sound source in this study. From the National Environment Agency (NEA) [56] report in Singapore, the maximum acceptable value of traffic noise level is about 67 dB averaged over an hour. However, people's exposure to noise levels above 65 dB can cause severe health problems [5]. Considering that the noise levels >65 dB could endanger the acoustic heath in urban environment [5], we assume that residents perceive traffic and construction noise annoyance level similarly at 65 dB. Eq 5 was applied to predict the percentage of noise acceptability for traffic noise.

To evaluate the percentage of environmental acceptability within a specific region, the area-weighted average factors ($\overline{PTA}$ and $\overline{PNA}$) over the pedestrian's way can be defined

$$\overline{PTA} = \frac{1}{A_{tot}} \int_{A_{tot}} PTA \, dS, \tag{6}$$

$$\overline{PNA} = \frac{1}{A_{tot}} \int_{A_{tot}} PNA \, dS, \tag{7}$$

where *dS* is the area differential element and $A_{tot}$ is the total area of the specific region. These two factors can facilitate summarizing the population the percentage of specific regions for urban planning and architectural design.

## 2.4 Study procedures

The procedures of this study can be summarized as shown in Fig 4. First, as discussed in Sec 2.1, the TMY weather data related to solar irradiance, the typical wind speed and direction from BCA, and the 3D STL model for the area of concern were set for input to conduct a

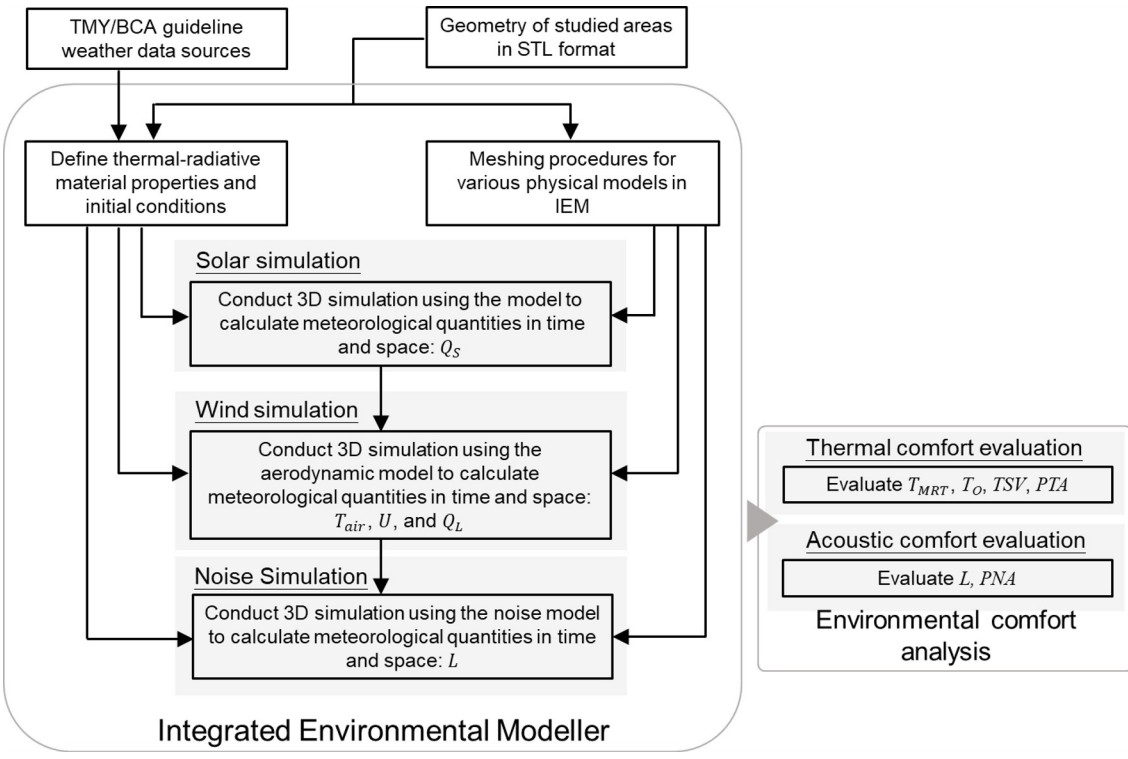

**Fig 4. The flowchart represents the detailed research procedures in this study.**

coupled simulation for the day and time of interest. Next, radiative and thermal material properties were assigned to each group of urban surfaces (concrete, grass, roads and water body) of the 3D STL model. IEM would then automatically generate meshes for the physical simulations. As discussed in Sec 2.2, the received shortwave irradiance ($Q_S$) mapping on the 3D model at a specific day and time was generated by the solar solver in IEM. Once $Q_S$ values on the surface of 3D model have been calculated, $Q_S$ will be taken into account in SEB model to determine the heat transfer boundary condition. Based on the heat transfer boundary condition, the meteorological quantities, air temperature ($T_{air}$), wind speed ($U$), and received longwave radiation ($Q_L$) were computed by the aerodynamic solver in IEM. The noise solver would calculate the noise level ($L$) in IEM, where $U$ is considered in the noise simulation. Finally, as discussed in Sec 2.3, the corresponding thermal indices ($T_O$, $T_{MRT}$, and $TSV$) and environmental comfort acceptability ($PTA$ and $PNA$) would be evaluated using Eqs 1–5, respectively.

## 3. Results and discussion

### 3.1 Three-dimensional coupled simulations of the urban district

The size of the studied district is 2518 m × 1762 m × 84 m. For solar and noise simulations, the district's extent in STL format defines the computational domain. The meshes were generated uniformly for the 3D geometries. The resolution of the mesh is 2 meters, and the total mesh number is about 2.1 million. In wind simulations, the size of the computational domain must be greater than the original size of the district for setting corresponding inlet atmospheric boundary conditions. As discussed in Sec 2.1, the inlet wind condition is assigned according to the 2019 [42] shown in Table 3. According to the wind directions at different times, the

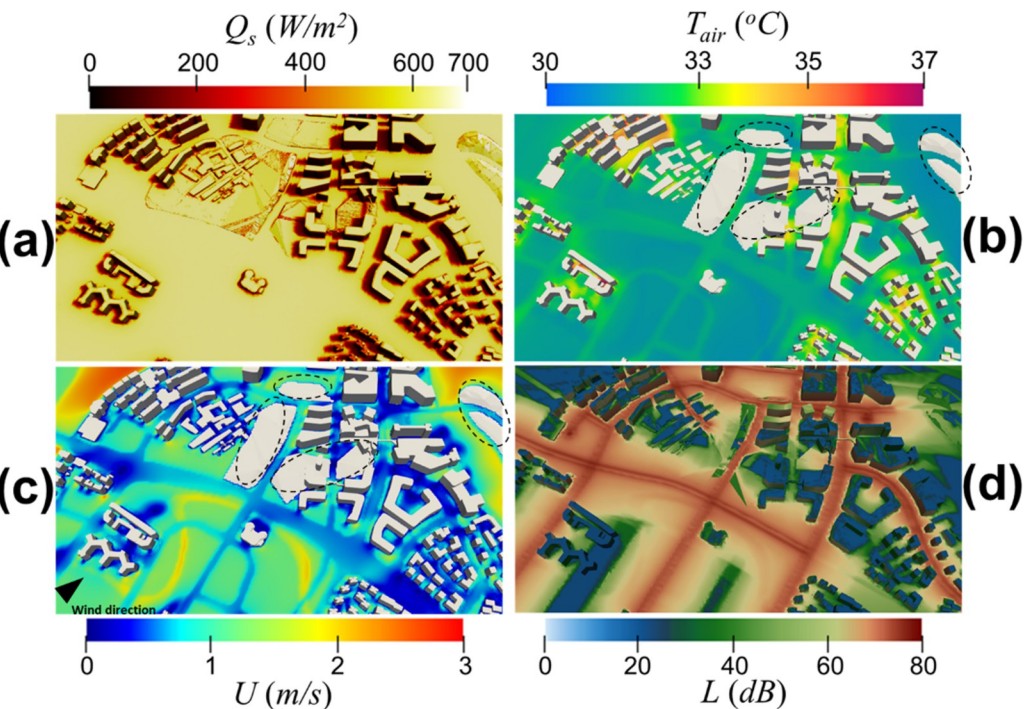

**Fig 5.** On the sunny day at 10:30, the figure shows **(a)** the solar irradiance mapping on the surface of the district, **(b)** air temperature as well as **(c)** wind speed contours at pedestrian level cut-plane (i.e., 1.5m) of the district, and **(d)** the noise level mapping on the surface of the district. The regions circled in **(b)** and **(c)** indicate that the terrain is higher than the 1.5m pedestrian level cut-plane. The arrow in **(c)** denotes the wind direction.

computational domain sizes are 44226 m × 30976 m × 605 m for the south inlet wind direction (i.e., the sunny day) and 22057 m × 30976 m × 605 m for the north-east inlet wind direction (i.e., the cloudy day). For generating the mesh for wind simulations, snappyHexMesh utility in OpenFOAM® has been used to discretize the computational domain. The total cell count of the computational domain is 17 million and consists of a maximum mesh resolution of 8 m away from the buildings and a minimum resolution of 2 m in the region of the building. The vertical resolution is 1 m. According to the Davenport-Wieringa roughness classification [57] and the BCA Green Mark standard [42], the aerodynamic roughness of inflow and ground surfaces were selected as 0.03 and 1.0, respectively. We adopted the initial isothermal conditions of 30.0°C air temperatures to carry out the simulations according to the BCA Green Mark standard [42].

In this section, we present physical quantities computed using IEM on the sunny day at 10:30 and on the cloudy day at 16:30, in which the solar radiation is the most direct dominant in each day. Fig 5 shows the simulated physical quantities on the sunny day at 10:30. Fig 5(A) is the solar irradiance mapping on the surfaces. The shadows of the buildings are on the west side due to the solar orientation in the morning. The air temperature and wind speed contours at the pedestrian level are shown in Fig 5(B) and 5(C), respectively. The wind profile (Fig 5(C)) is determined by the inflow wind direction, topography, and material properties. It can be seen in Fig 5(B) that the hot spots are in the regions with abundant solar radiation ($Q_s >$ 600W/m$^2$) and low wind speed ($U < 0.5$m/s). Fig 5(D) illustrates the 3D traffic noise map of the district. According to the traffic noise sources defined in Fig 1(A) and Table 2, the color in the middle of the road is significantly darker, implying the roads have a higher noise level. This noise mapping at the district comes from calculating the noise propagation from the

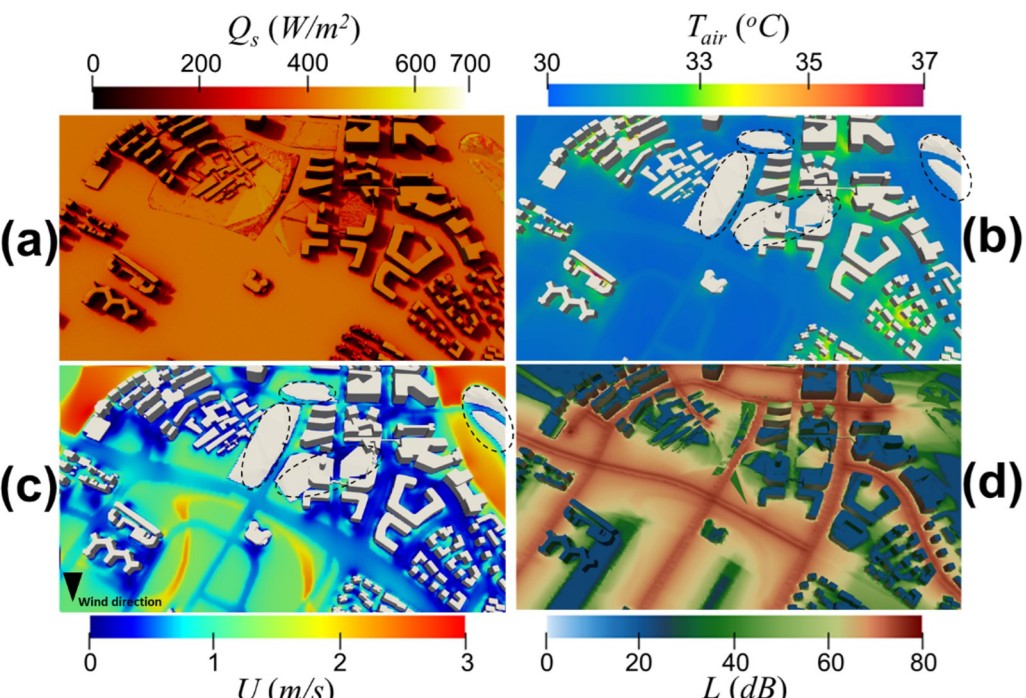

**Fig 6.** On the cloudy day at 16:30, the figure shows **(a)** the solar irradiance mapping on the surface of the district, **(b)** air temperature as well as **(c)** wind speed contours at pedestrian level (i.e., 1.5-m) of the district, and **(d)** the noise level mapping on the surface of the district. The regions circled in **(b)** and **(c)** indicate that the terrain is higher than the 1.5m pedestrian level cut-plane. The arrow in **(c)** denotes the wind direction.

source and the noise reduction due to obstruction, distance attenuation, ground absorption, and noise barrier screen. With such noise mapping and Eq 5, we can further evaluate the acoustic comfort in areas with high pedestrian traffic during peak hours and discuss this in Sec 3.2.

On the other hand, Fig 6 shows the simulated physical quantities on the cloudy day at 16:30. The shadows of the buildings are on the east side due to the solar orientation in the afternoon. Fig 6(A) also shows that the amplitude of overall solar irradiance is relatively weak compared with that shown in Fig 5(A). It is reflected in the air temperature profile shown in Fig 6(B), where a decrease in global solar radiation decreases the overall air temperature. Under the different inflow directions, the wind speed profile at that time is revealed in Fig 6 (C). Fig 6(D) shows the corresponding noise mapping. There is no noticeable difference between Figs 5(D) and 6(D) because the temperature gradient and wind shear impact on traffic noise propagation of both cases is relatively low for the area and tropical climate under study.

## 3.2 Evaluation and analysis of thermal comfort on the study sites

To further evaluate the environmental comfort in the area with high pedestrian traffic are marked in Fig 1(B), Fig 7 shows the thermal comfort index (i.e., *TSV*) and acoustic comfort index (i.e., *L*) contours at the pedestrian level (i.e., 1.5 m). The *TSV* results are calculated using Eq 3 based on the simulation results shown in Figs 5(A)–5(C) and 6(A)–6(C). Fig 7(A) shows that *TSV* >2 in more than 80% of the area. The outdoor environment is considered hot on the sunny day at 10:30. In contrast, *TSV* on a cloudy day at 16:30, shown in Fig 7(C), is relatively low (<1) for >60% of the area of interest. The high *TSV* generally happens due to high solar irradiance, and due to the weak wind flow that is usually associated with a low wind speed

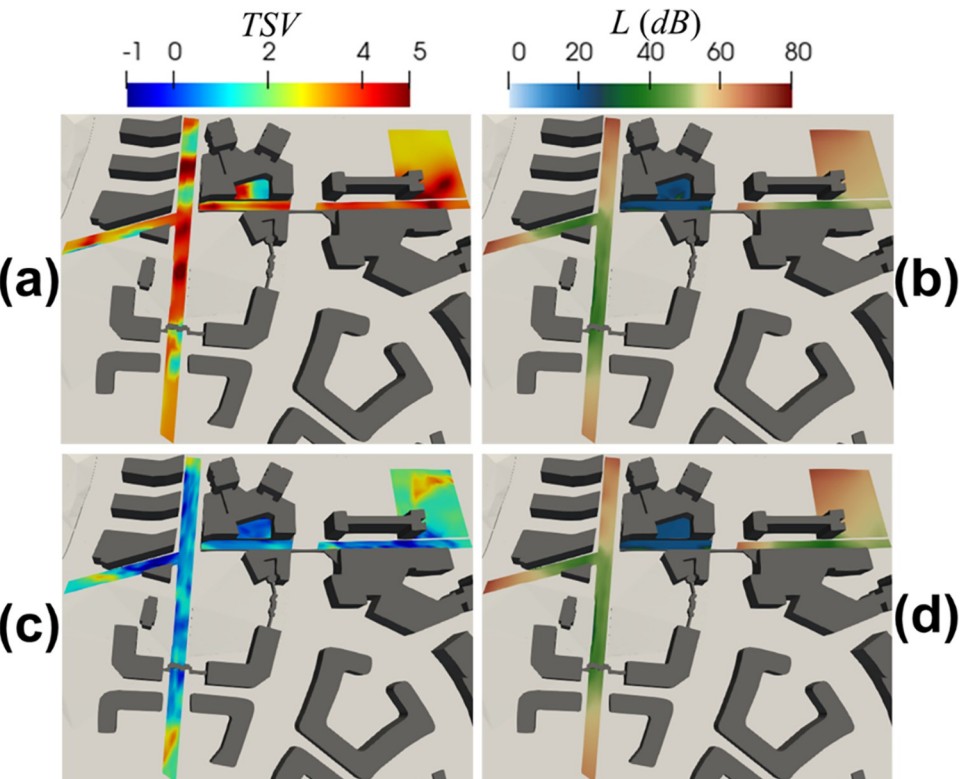

**Fig 7.** According to the simulation results shown in Figs 5 and 6, the figure shows contours of comfort indices of **(a)** *TSV* and **(b)** noise level at the sunny day 10:30, and **(c)** *TSV* and **(d)** noise level at the cloudy day 16:30.

of $\leq$ 2 m/s (see Fig 6(C)). Thus, in Fig 7(A) and 7(C), *TSV* hotspots occur in areas with great solar exposure and inadequate natural ventilation, strongly affected by street geometry, building orientation, wind direction, and sun position. For example, in Fig 7(A), the hotspot can be observed in the south area of the site (iii), where it is unshaded and lacks the speed for adequate permeability of natural ventilation on the sunny day 10:30. In contrast, in Fig 7(C), the reduction of *TSV* in the same place is because solar irradiance, sun position, and wind direction change on the cloudy day 16:30. The contour of the noise level (*L*) shown in Fig 7(B) and 7(D) are similar, indicating that the effect of wind and thermal shear are weak for the area of interest. These results in Fig 7(B) and 7(D) are derived from the IEM outputs as shown in Figs 5(D) and 6(D).

Environmental acceptability as a people-centric way to evaluate thermal comfort for architecture and urban design, compared to using typical thermal comfort metrics (such as *TSV*) is also evaluated here. Fig 8(A) shows the relationship between the percentage of thermal acceptability (*PTA*) and *TSV* calculated following Eq 4. Similarly, Fig 8(B) shows the relationship between the percentage of noise acceptability (*PNA*) and noise level according to Eq 5. The thresholds for thermal and acoustic comfort metrics are defined as 50% of the population feeling acceptable at *TSV* = 2.3 and *L* = 62.6 dB, respectively. The qualified evaluation of environmental comfort at a specific location is achievable based on whether the *PTA* and *PNA* are over 0.5. It is noted that the threshold (*PTA* = 0.5 and *PNA* = 0.5) used in this study is an estimate of the feelings of the simple majority rather than the universal definition. From different studies [58–60], other environmental comfort thresholds were introduced based on different environmental matrices for the specific purpose.

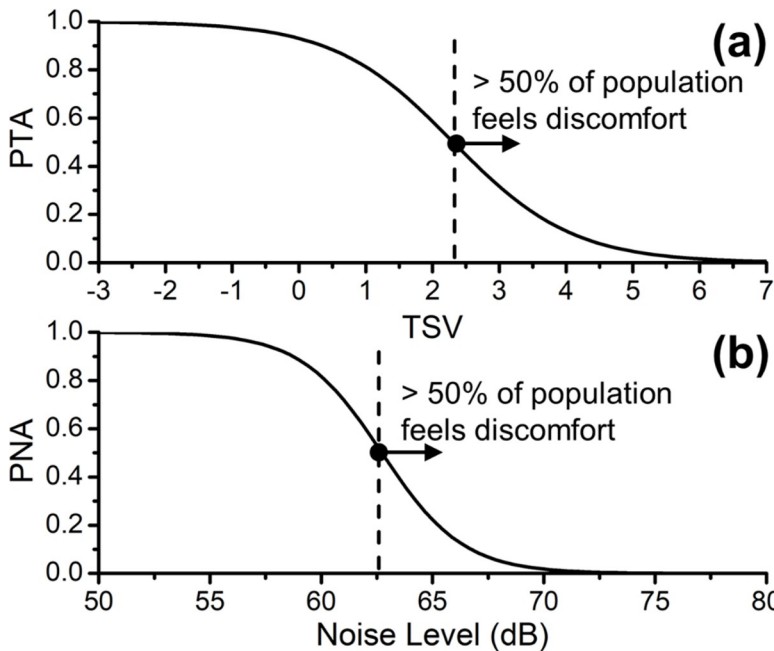

**Fig 8.** Shows **(a)** the relationship between the percentage of thermal acceptability (*PTA*) and thermal sensation vote (*TSV*) obtained from Eq 4, and **(b)** the relationship between the percentage of noise acceptability (*PNA*) and noise level obtained from Eq 5, respectively. The dashed lines denote the index threshold (*PTA* = 0.5 and *PNA* = 0.5) of that over 50% population feels discomfort (i.e., the right area pointed by the arrow).

From Figs 9–12, we evaluate the environmental comfort at each site (i, ii, iii, and iv) within the area with high pedestrian traffic are marked in Fig 1(A). Fig 9(A) shows the area-averaged *PTA* (i.e., $\overline{PTA}$) of the sunny day and the cloudy day on the east-west boulevard. The cloudy day has higher $\overline{PTA}$s than the sunny day during the daytime, consistent with the trend of solar irradiance shown in Fig 2. The error bars illustrate the huge variation between local maximum and minimum *PTA* at different locations along the campus boulevard. The local variation of *PTA* strongly depends on the extent of the shade projected by the surrounding buildings at that hour of interest. There are two points in time (10:30 and 13:30) that $\overline{PTA}$ is below 0.5 on the sunny day and the corresponding minimum *PTA* is very close to 0. We expect these two points to be under a relatively high global solar irradiance, and >50% of people are uncomfortable with the thermal environment. In contrast, Fig 9(B) shows that the area-averaged *PNA* (*i.e.* $\overline{PNA}$) is almost always close to a constant (~0.9) that far exceeds the baseline ($\overline{PNA}$ = 0.5) all the time during peak hours, even though the local minimum *PNA* is close to 0. In Fig 9(C), the spatial distribution of *PTA* and *PNA* at 13:30 on the sunny day (worst case) is illustrated. It is obvious that most of the area along the campus boulevard is expected to be thermally uncomfortable with *PTA*<0.5 (red and purple zones), and the spaces, where *PNA*<0.5, are concentrated near the two road junctions (i.e., traffic noise sources defined in Fig 1(A)), which overlaps with a part of the *PTA*<0.5 region, marked in purple. The small zones with, *PTA*<0.5 and *PNA*>0.5; and; *PTA*>0.5 and *PNA*>0.5 along the bottom edge of the purple zone with *PTA*<0.5 are marked in amber and green, respectively. This indicates that traffic noise level is always acceptable by most people even at peak hour traffic except for people who stay in the small region directly affected by the traffic noise. Local noise mitigation measures are required in such the areas if open air retail spaces are planned in the purple and amber regions, where *PTA*<0.5.

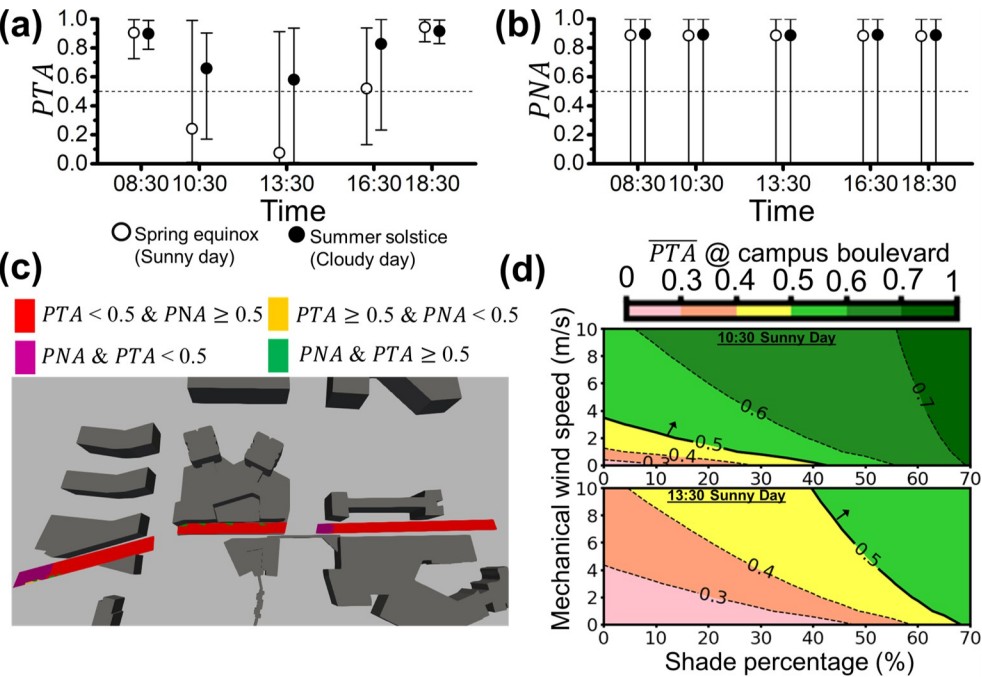

**Fig 9.** The area-averaged **(a)** *PTA* and **(b)** *PNA* of the campus boulevard (site (i)) for both sunny (open circles) and cloudy (filled circles) days are represented with open and filled circles (i.e., $\overline{PTA}$ and $\overline{PNA}$), respectively. The error bars extended from the area-averaged *PTA* or *PNA* (circles) represent the local maximum and local minimum *PTA* or *PNA* on the campus boulevard. The environmental acceptability threshold of 0.5 is denoted by the dotted line (i.e., *PTA* = 0.5 or *PNA* = 0.5). **(c)** The spatial plot of *PTA* and *PNA* above or below the threshold of 0.5 for campus boulevard at 13:30 on the sunny day (worst case) is illustrated. **(d)** The contour plot shows the potential improvement of $\overline{PTA}$ with the increase in shade and/or wind speed at 10:30 and 13:30 on the sunny day for the campus boulevard.

For improving overall environmental comforts on the specific site for high-level retail planning, the averaged indicators (i.e., $\overline{PTA}$ and $\overline{PNA}$) rather than local values (i.e., *PTA* and *PNA*) will be further analyzed. As compared to $\overline{PNA}$ shown in Fig 9(B), the improvement of $\overline{PTA}$ is the primary objective to be achieved by architectural solutions. We provide a simplified parametric analysis to improve thermal comfort at the hot times while $\overline{PTA}$ <0.5 (i.e., 10:30 and 13:30 of the sunny day) by decreasing solar irradiance and increasing the wind speed from the simulation results, and other conditions held constant. Tuning the two parameters may be done using setting up shelters to block the incoming solar radiation and adding ventilation fans to increase the local wind speed under ideal conditions. The contour plot in Fig 9(D) shows the results of the improvement of $\overline{PTA}$ by decreasing the solar irradiance and increasing the wind speed. Increasing shade percentage (i.e., decreasing solar irradiance) and increasing wind speed are set from 0 to 70% and 0 to 10 m/s, respectively. The increasing wind speed of 10 m/s is the maximum boost provided by typical outdoor ventilation fans [61]. The solid lines in Fig 9(D) indicate the threshold of $\overline{PTA}$ = 0.5 to separate the region with $\overline{PTA}$ >0.5 (green colors) and the region with $\overline{PTA}$ <0.5 (warm colors). The intersections of the solid line and vertical and horizontal axes represent the minimum requirements for the adopting the single way (only increasing wind speed and only increasing shade percentage) to achieve $\overline{PTA}$ = 0.5. At 10:30 of the sunny day (the upper panel of Fig 9(D)), $\overline{PTA}$ = 0.5 can be achieved by increasing 3.7 m/s in wind speed or blocking 42% of solar irradiance. The corresponding value of wind speed and shade percentage on the solid line also denotes the minimum coupling mechanism criterion to achieve $\overline{PTA}$ = 0.5. For instance, we can improve $\overline{PTA}$ to 0.5 by increasing 2

m/s in wind speed and blocking 15% of solar irradiance simultaneously. In the lower panel of Fig 9(D), it is observed that the further increase in wind speed and shade percentage is required to reach the $\overline{PTA}$ baseline under a higher global solar irradiance at 13:30 of the sunny day as compared to that at any other time. $\overline{PTA}$ = 0.5 can be achieved by blocking 68% of solar irradiance but cannot be achieved by only increasing the wind speed of <10 m/s.

Fig 10(A) and 10(B) show the area-averaged $\overline{PTA}$ and $\overline{PNA}$ of the sunny day and the cloudy day along the heritage trail. Unlike Figs 9(A) and 10(A) shows that $\overline{PTA}$s are less than 0.5 with the corresponding minimum $PTA \cong 0$ at 10:30 and 13:30 of the cloudy day. In Fig 10(B), $\overline{PNA}$s are between 0.84 and 0.88, slightly less than that shown in Fig 9(B), but the corresponding minimum $PNA$s are close to 0 all the time. Fig 10(C) shows that most of the area of the heritage trail is exposed to the thermal problem with $PTA<0.5$ (red and purple zones) at 13:30 on sunny day. At the same time, the space overlapped the area with $PNA<0.5$ (purple zones) is located at the part of the north end of the heritage trail. The green ($PTA<0.5$ and $PNA>0.5$) and amber ($PTA>0.5$ and $PNA>0.5$) zones are almost nonexistent. Fig 10(C) and 10(D) show the improvement of $\overline{PTA}$ via two ways on the sunny and cloudy day. At 10:30 of the sunny day (the upper panel of Fig 10(C)), using increasing 2.6 m/s in wind speed or blocking 32% of solar irradiance can reach $\overline{PTA}$ = 0.5. As can be seen in the lower panel of Fig 10(C), $\overline{PTA}$ = 0.5 can be achieved by blocking at least 68% of the incoming solar irradiance at 13:30 of the cloudy day, but the same $\overline{PTA}$ value cannot be achieved by only increasing the wind speed of up to 10 m/s. On the other hand, as shown in Fig 10(D), we can improve $\overline{PTA}$ to reach 0.5 via adding 6.7% shade percentage or increasing 0.3 m/s in wind speed at 10:30 of the cloudy day. $\overline{PTA}$ = 0.5 is achievable by adding 21% shade percentage or increasing 1.1 m/s in wind speed at 13:30 of the cloudy day.

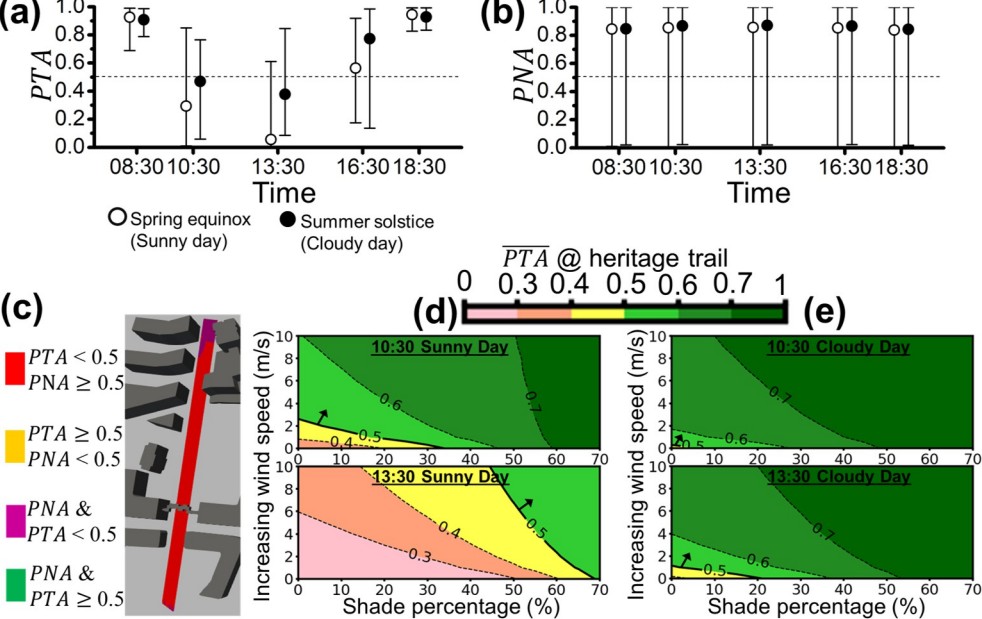

**Fig 10.** The area-averaged **(a)** *PTA* and **(b)** *PNA* of the heritage trail (site (ii)) for both sunny (open circles) and cloudy (filled circles) days are represented with open and filled circles (i.e., $\overline{PTA}$ and $\overline{PNA}$), respectively. The error bars extended from the area-averaged *PTA* or *PNA* (circles) represent the local maximum and minimum *PTA* or *PNA* on the heritage trail. The environmental acceptability threshold of 0.5 is denoted by the dotted line (i.e., *PTA* = 0.5 or *PNA* = 0.5). **(c)** The spatial plot of *PTA* and *PNA* above or below the threshold of 0.5 for heritage trail at 13:30 on the sunny day (worst case) is illustrated. The contour plot shows the potential improvement of $\overline{PTA}$ with the increase in shade and/or wind speed at 10:30 and 13:30 on **(d)** the sunny day, and **(e)** the cloudy day for the heritage trail.

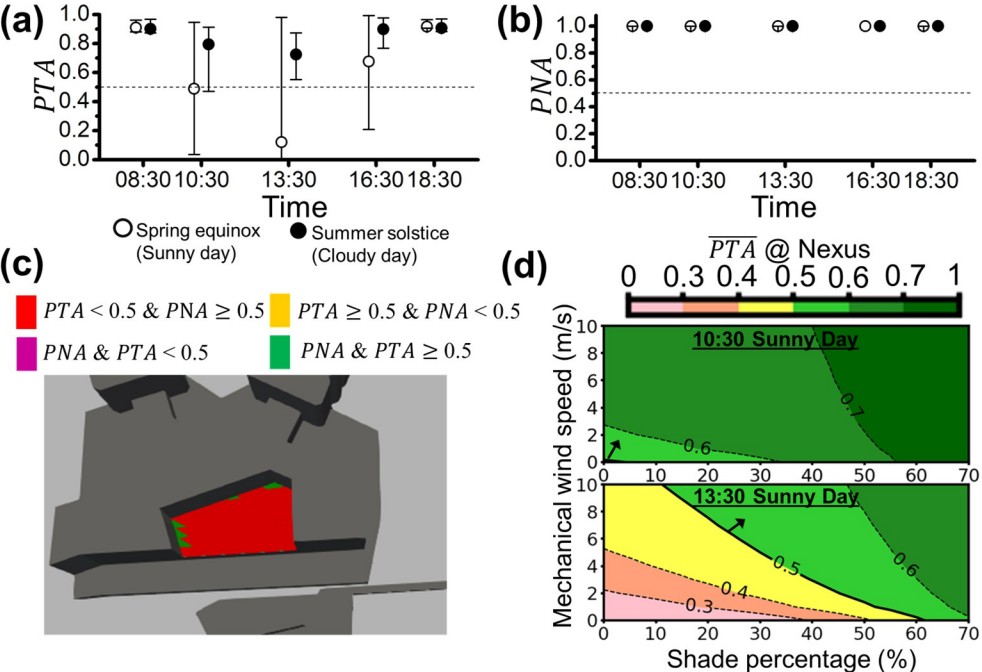

**Fig 11.** The area-averaged **(a)** *PTA* and **(b)** *PNA* of Nexus (site (iii)) for both sunny (open circles) and cloudy (filled circles) days are represented with open and filled circles (i.e., $\overline{PTA}$ and $\overline{PNA}$), respectively. The error bars extended from the area-averaged *PTA* or *PNA* (circles) represent the local maximum and local minimum *PTA* or *PNA* on Nexus. The environmental acceptability threshold of 0.5 is denoted by the dotted line (i.e., *PTA* = 0.5 or *PNA* = 0.5). **(c)** The spatial plot of *PTA* and *PNA* above or below the threshold of 0.5 for Nexus at 13:30 on the sunny day (worst case) is illustrated. **(d)** The contour plot shows the potential improvement of $\overline{PTA}$ with the increase in shade and/or wind speed at 10:30 and 13:30 on the sunny day for Nexus.

Fig 11(A) and 11(B) show the area-averaged $\overline{PTA}$ and $\overline{PNA}$ of the sunny day and the cloudy day at the Nexus. In Fig 11(A), $\overline{PTA}$s are higher than that shown in Figs 9(A) and 10(A), especially on the cloudy day. However, the minimum *PTA* is still close to 0 on the sunny day, e.g., at 13:30 (worst case). From Fig 11(B), $\overline{PNA}$ is always close to 1 with almost zero variation that 100% of people satisfy the acoustic environment all the time during peak hours. The spatial feature of *PTA* and *PNA* at 13:30 on the sunny day as shown in Fig 11(C) can be seen that the most of area is with *PTA*<0.5 marked in red, few scattered areas with *PTA*≥0.5 are marked in green, and almost no area is with *PNA*<0.5. Fig 11(D) shows the improvement of $\overline{PTA}$ via two strategies on the sunny day. At 10:30 of the sunny day (the upper panel of Fig 11(D)), merely using increasing 0.2 m/s in wind speed or blocking 4.2% of solar irradiance can reach $\overline{PTA}$ = 0.5. In the lower panel of Fig 11(D), the baseline of $\overline{PTA}$ can be achievable by blocking 62% of solar irradiance at 13:30 of the sunny day but still be unachievable by solely increasing the wind speed.

Fig 12(A) and 12(B) show the area-averaged $\overline{PTA}$ and $\overline{PNA}$ with the corresponding variation of the sunny day and the cloudy day at the market village. Similar to the results (see Fig 9(A)), $\overline{PTA}$s are less than 0.5 and the corresponding minimum *PTA*s are close to 0 at 10:30 and 13:30 of the sunny and cloudy day as shown in Fig 12(A). It can be observed that the maximum *PTA*s at 10:30 and 13:30 of the sunny and cloudy day are much less than those elsewhere. On the other, in Fig 12(B), $\overline{PNA}$s are between 0.69 and 0.73, slightly less than that shown in the results from Figs 9(B), 10(B) and 11(B). The corresponding minimum *PNA*s are close to 0 all

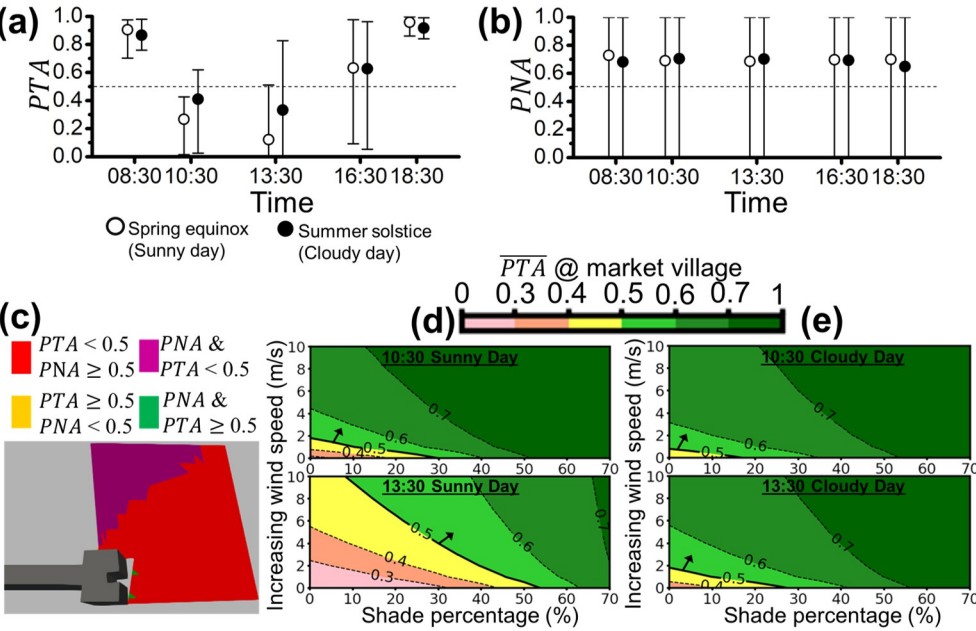

**Fig 12.** The area-averaged **(a)** *PTA* and **(b)** *PNA* of the market village (site (iv)) for both sunny (open circles) and cloudy (filled circles) days are represented with open and filled circles (i.e., $\overline{PTA}$ and $\overline{PNA}$), respectively. The error bars extended from the area-averaged *PTA* or *PNA* (circles) represent the local maximum and minimum *PTA* or *PNA* on the campus boulevard. The environmental acceptability threshold of 0.5 is denoted by the dotted line (i.e., *PTA* = 0.5 or *PNA* = 0.5). **(c)** The spatial plot of *PTA* and *PNA* above or below the threshold of 0.5 for market village at 13:30 on the sunny day (worst case) is illustrated. The contour plot shows the potential improvement of $\overline{PTA}$ with the increase in shade and/or wind speed at 10:30 and 13:30 on **(d)** the sunny day, and **(e)** the cloudy day for the market village.

the time. Fig 12(C) shows that the zone with *PTA*<0.5 (red and purple) is almost everywhere and a part of space overlapped with *PNA*<0.5 (purple) appears beside the noise sources. Fig 12 (D) and 12(E) show the improvement of $\overline{PTA}$ via two strategies on the sunny and cloudy day. At 10:30 of the sunny day (the upper panel of Fig 10(D)), using increasing 1.8 m/s in wind speed or blocking 29% of solar irradiance, we can achieve $\overline{PTA}$ = 0.5. In the lower panel of Fig 12(D), $\overline{PTA}$ = 0.5 can be achieved at 13:30 of the sunny day by blocking 54% of solar irradiance. On the other hand, as shown in Fig 10(E), we can improve $\overline{PTA}$ to reach 0.5 via adding a 17% shade percentage or increasing 0.8-m/s in wind speed at 10:30 of the cloudy day. $\overline{PTA}$ =

**Table 4. The table summarizes the conditions to achieve** $\overline{PTA}$ **= 0.5 from Figs 9(D), 10(D), 10(E), 11(D), 12(D) and 12(E).** In each cell, the upper panel is the required shade percentage, and the upper panel is the required increasing wind speed. The cells in green represent that $\overline{PTA} \geq 0.5$ can be naturally achieved. The cells in red represent that cannot be achieved by only increasing the wind speed in a reasonable range of up to 10 m/s.

| | | campus boulevard site (i) | heritage trail site (ii) | Nexus site (iii) | market village site (iv) |
|---|---|---|---|---|---|
| sunny day | 10:30 | 42% | 32% | | 29% |
| | | 3.7 m/s | 2.6 m/s | | 1.8 m/s |
| | 13:30 | 68% | 68% | 4.2% | 54% |
| | | >10 m/s | >10 m/s | 0.2 m/s | >10 m/s |
| cloudy day | 10:30 | | 6.7% | | 17% |
| | | | 0.3 m/s | | 0.8 m/s |
| | 13:30 | | 21% | | 27.5% |
| | | | 1.1 m/s | | 1.8 m/s |

0.5 is achievable via adding 27.5% shade percentage or increasing 1.8-m/s wind speed at 13:30 of the cloudy day. The overall conditions to achieve $\overline{PTA}$ = 0.5 from Figs 9(D), 10(D), 10(E), 11(D), 12(D) and 12(E) are summarized in Table 4.

### 3.3 Discussion

From Figs 9–12, $\overline{PTA}$ and $\overline{PNA}$ directly represent the overall human acceptance of outdoor environmental conditions in a specific area. The variation of *PTA* and *PNA* are affected by the local environmental conditions. At each time, the maximum or minimum *PTA* and *PNA* represent the most acceptable or unacceptable extent locally for thermal and acoustic comfort in different studied sites, respectively. The results (Figs 9(B), 10(B), 11(B) and 12(B)) show that $\overline{PNA}$s do vary slightly (between 0.69 and 1) for the five different time slots due to the different wind and air temperature conditions for all the four sites in the business district. The site with a greater number of surrounding buildings, which act as acoustic barriers, has a higher *PNA*. For instance, the site (iii) (Nexus) has the highest $\overline{PNA}$ (= 1) among four sites and the site (iv) (market village) has the lowest $\overline{PNA}$ (~ 0.7). Although $\overline{PNA}$s is in the range between 0.7 and 1, the corresponding minimum *PNA* is close to 0 except for site (iii) (Nexus). From the results (Figs 9(C), 10(C) and 12(C)), the local zones with a small *PNA* are always near the noise sources. Among four sites in this study, the north-west region of site (iv) is exposed to the most serious traffic noise because this region is surrounded by noise sources without any acoustic barriers, like buildings. The straightforward way to improve the acoustic comfort by constructing local acoustic barriers to protect pedestrians from the traffic noise. However, the acoustic barriers will block airflow transport, further endangering thermal comfort. Thus, outdoor retail activities would not be recommended in the purple zones (*PNA*<0.5 and *PTA*<0.5) as shown in Figs 9(C), 10(C) and 12(C). For the four study sites, *PNA* < 0.5 is found in the small area near the traffic noise sources. *PNA* is always greater than 0.5 for most of the spaces in the four sites. In contrast, *PTA* < 0.5 for most of the spaces is always observed in the worst-case scenario as shown in Figs 9(C), 10(C), 11(C) and 12(C). For high-level retail planning, $\overline{PTA}$ and $\overline{PNA}$ are representative metrics. $\overline{PNA}$ is always far beyond 0.5 in each site, which means no immediate action is required to mitigate traffic noise if the retail places are not near the planned roads in the business district. Considering Singapore's relatively low wind speed (< 3 m/s), solar irradiance becomes the essential parameter affecting $\overline{PTA}$. The results (Figs 9(A), 10(A), 11(A) and 12(A)) show a high correlation between $\overline{PTA}$ and solar irradiance, as shown in Fig 2. As such, $\overline{PTA}$ is observed reaching the minimum at noon of both days, where the global solar irradiance is the maximum each day. To ensure that $\overline{PTA}$ can at least reach the baseline of 0.5 all the time, we evaluate the improving $\overline{PTA}$ at the hot times (Figs 9(D), 10(D), 10(E), 11(D), 12(D) and 12(E)) by increasing wind speed, blocking solar irradiance, or adopting above both simultaneously. In these contour plots, the solid line separates the parametric space to achieve $\overline{PTA} \geq 0.5$ (i.e., the green-colored region pointed by the arrow in contour plots) and $\overline{PTA} < 0.5$ (i.e., the warm-colored region). The intersections of the solid lines and vertical axis and horizontal axes represent the minimum requirements for adopting the only increasing wind speed and only blocking solar irradiance to achieve $\overline{PTA}$ = 0.5. The parametric space of the required conditions to achieve $\overline{PTA}$ = 0.5 at 13:30 (the lower panel of Figs 9(D), 10(D), 11(D) and 12(D)) of the sunny day is much smaller than that at any other time (the upper panel of Figs 9(D), 10(D), 10(E), 11(D), 12(D) and 12(E)). The corresponding required conditions for achieving $\overline{PTA} \geq 0.5$ at each site at 13:30 (the worst-case scenario) can be applied to achieve $\overline{PTA} \geq 0.5$ at any other time. For example, in the lower panel

of Fig 10(D), the requirements for achieving $\overline{PTA} \geq 0.5$ also satisfy that as shown in the upper panel of Fig 10(D) and 10(E).

If considering using the single strategy for improving $\overline{PTA}$ at the hottest time (i.e., 13:30 of the sunny day as shown in the lower panel of Figs 9(D), 10(D), 11(D) and 12(D), blocking the solar irradiance is an efficient way to achieve $\overline{PTA}$ = 0.5 when the solar radiation is the direct dominant with a small zenith angle. Although the layout of surrounding buildings is different at study sites, the improvement trends of $\overline{PTA}$ are similar, and $\overline{PTA}$ at all four sites can be improved from <0.2 to 0.5 by blocking 54–68% of solar irradiance, respectively. It should be noted that $\overline{PTA}$ = 0.5 at the site (iv) (market village) can be achieved by blocking 54%, which is the minimum among the four sites. This fact means that blocking solar irradiance is more efficient to improve $\overline{PTA}$ in open spaces that agree with the findings from [62–65]. Increasing wind speed is another way to improve $\overline{PTA}$. However, the results at 13:30 of the sunny day (i.e., the lower panel of Figs 9(D), 10(D), 11(D) and 12(D)) show that $\overline{PTA}$ is always less than 0.5 even though the wind speed is increased by 10 m/s, which is the typical maximum wind speed for mechanical ventilation [61]. In contrast, the results at any other times when 0.3< $\overline{PTA}$ <0.5 show that $\overline{PTA} \geq$0.5 is achievable by increasing the wind speed to 0.2–3.7 m/s. As stated above, we can deduce that blocking solar irradiance can efficiently improve $\overline{PTA}$ at any time on both days, especially when $\overline{PTA}$ <0.2 (at 13:30 of the sunny day). On the other hand, enhancing ventilation by increasing wind speed is effective at the time when $\overline{PTA}$ >0.3 (e.g., the cloudy day) in the first place.

Furthermore, the combined effects of blocking solar irradiance and increasing wind speed can be a more efficient way to improve $\overline{PTA}$ than just adopting shade or boosting the wind. In the contour plots from Figs 9–12, every point on the solid line represents a combination of conditions to reach $\overline{PTA}$ = 0.5. For instance, the lower panel of Figs 9(D), 10(D), 11(D) and 12 (D) show that the required blocking percentage of solar irradiance can be decreased ~10% for reaching $\overline{PTA} \geq$0.5 at 13:30 of the sunny day when the wind speed is increased by 2 m/s which is roughly conventional fan airspeed driven by ceiling fans [66]. In practicality, this way provides flexibility in architecture and has been widely used around al fresco restaurants, outdoor amphitheaters, and some sheltered pedestrian ways in Singapore.

The main implication of the result is that $\overline{PTA}$ and $\overline{PNA}$ give an indication on the percentage of visitors who find the location comfortable enough for them to patronize or hang around al fresco dining and pop-up retail kiosks in the outdoor environment. In this study, $\overline{PTA}$ is a dominant factor to determine people's willingness to retail queue or gather at all four sites at the hot time. On the other hand, although the $\overline{PNA}$ is always >0.5 at each site, traffic noise issue in the area with $PNA$<0.5 still should be considered in the retail planning. From the spatial color mapping plots (Figs 9(D), 10(D), 11(D) and 12(D)), the regions with $PNA$<0.5 are always associated with $PTA$<0.5, marked in purple. This clearly defines the zone that is not recommended for outdoor retails if the thermal and acoustic comfort cannot be improved simultaneously. According to the results, specific guidelines for retail planning for each site can be made as follows. The site (i) (campus boulevard) is a thoroughfare with buildings on either side. The $\overline{PTA}$ along this thoroughfare can be improved by installing pop-up shopping areas next to the side buildings and awnings for visitor queuing. To increase local airspeed using outdoor ventilation devices will further improve $\overline{PTA}$. In comparison between the results, Figs 9(A) and 10(A), site (ii) (heritage trail) is hotter than site (i) due to the lack of buildings on the side to block the solar irradiance. The building shadows can directly improve the thermal comfort [67, 68]. The site (ii) is planned to be a pedestrian-only pathway renovated

from an old road. Thus, $\overline{PTA}$ here can be expected to improve by installing a sheltered walkway [69] and planting street trees with overhead coverage [55, 70, 71]. The location of retail shops or kiosk that may attract long queues are not recommended in the purple zones (Figs 9(C) and 10(C)) near the road junctions at the site (i) and (ii). The site (iii) (Nexus) has a higher $\overline{PTA}$ benefiting from the spontaneous shading by three-sides surrounding buildings. With sufficient shading percentage by the shelter and increased wind speed via ceiling fans suggested by Fig 11(D), the tenants or visitors are expected to be more willing to stay in the outdoor area for al fresco dining [72] and pop-up shopping over a long time. In addition, site (iii) is the only one for outdoor activities free from interfering traffic noise in this study as seen in Fig 11(C). The site (iv) (market village) is an open area for outdoor activities. According to the results from Fig 12(D), $\overline{PTA}$ can be efficiently improved by blocking solar irradiance via installing the canopy frames and planting trees. Keeping away from the area affected by traffic noise, the purple zone in Fig 12(C), outdoor amphitheaters with ceiling fans have a suitable setting for outdoor events.

If the shading techniques and mechanical wind devices may be challenging to deploy practically to reach the condition for $\overline{PTA}$ improvement, people may tend to remain indoors by relying on air conditioning during the hot time. In addition, outdoor retail activities might possibly affect people's health in the area with a high-level thermal and noise exposure, i.e., purple zones in Figs 9(C), 10(C) and 12(C). Because of that, urban planners and architects can choose retail options at outdoor locations for specific purposes. We have pointed out that the threshold of $\overline{PTA}$ = 0.5 and $\overline{PNA}$ = 0.5 is the baseline used in this study but not universal. The policy of the $\overline{PTA}$ and $\overline{PNA}$ standards can be formulated depending on the actual circumstances, e.g., $\overline{PTA} \geq 0.6$. According to Table 4 (the results from Figs 9(D), 10(D), 10(E), 11(D), 12(D) and 12(E)), the architects and urban planners can find the corresponding conditions for improving $\overline{PTA}$ locally.

We note that this study had only examined the potential improvement on $\overline{PTA}$ via the simplified parametric analysis (i.e., decreasing the solar irradiance and increasing the wind speed) under ideal conditions, without considering trees, shades, or outdoor fans in the simulations. Under the ideal conditions, we assumed that the solar radiation level and wind speed have implicitly incorporated the effects of trees, shades, or outdoor fans. The double parametric space in terms of solar radiation and wind speed provides a guideline for architects to quickly identify the optimal microclimate conditions for planning the infrastructure and landscape combination during the scenario design stage. The exact details of the surrounding environment, i.e., shades, trees and fans, can be inversely designed based on the optimal microclimate condition, which is out of the scope of this paper and requires further study. Although the evaluation provides the parametric scope and the options to improve outdoor thermal comfort for future plan of combining landscape and infrastructure (e.g., trees with shelter walkways, green walls with outdoor ventilation fans, and etc.) in design, the solar-wind-coupled simulations and on-site measurements with regards the as-built remains to be done at a later stage.

## 4. Conclusions

In conclusion, this paper evaluates the outdoor environmental comfort of an emerging district in Singapore. This evaluation was based on Integrated Environmental Modeler's (IEM) simulation results, a 3D coupled physics simulation platform that includes solar radiation, wind, thermal, and traffic noise, using a single unified set of geometrical input. From the results of the IEM, we further derived environmental indices (i.e., *TSV* and *L*) on the four studied sites with high pedestrian traffic. These four sites are either pedestrian thoroughfares, or gathering

areas for potential outdoor community events and activities for the tenants and residents staying near the business district. The people-centric acceptability factors (i.e., *PTA* and *PNA*), defined as a function of *TSV* and *L*, were introduced to define the degrees of environmental comfort. The variance of *PTA* and *PNA* come from the spatial distribution of environmental conditions in this business district. In the worst-case scenario, i.e., at 13:30 on the sunny day, *PTA*<0.5 is observed almost everywhere along all sites. In the meantime, the places, *PNA*<0.5, are found concentrated near the noise sources, which overlaps with a part of the *PTA*<0.5 region. These areas (purple zones in Figs 9(C), 10(C) and 12(C)) are not recommended for outdoor activities for retail purposes because improving *PNA* by constructing the noise barriers may further degrade the local *PTA*. For the high-level retail planning, we set the acceptable criterion of $\overline{PTA}$ = 0.5 and $\overline{PNA}$ = 0.5, meaning most people would be satisfied with the thermal and acoustic environment, respectively. $\overline{PNA}$ is always far beyond 0.5 in the studied sites, which means that most people here are satisfied with the acoustic comfort influenced by the proximity and road categories in the area except for people stay in purple zones as shown in Figs 9(C), 10(C) and 12(C). In contrast, $\overline{PTA}$ is highly dependent on global solar irradiance due to the low-wind-speed conditions which is always observed < 3 m/s in this study. Thus, $\overline{PTA}$ trends towards the minimum at noon of the sunny day when global solar irradiance is maximum. We proposed a simplified parametric analysis to improve the $\overline{PTA}$ when $\overline{PTA}$ < 0.5, (1) by blocking the solar irradiance and (2) by increasing the wind speed. The improvement of $\overline{PTA}$ is a function of the shading percentage and the mechanical wind speed. Blocking solar irradiance is effective to improve $\overline{PTA}$ at 13:30 of the sunny day, i.e., the hottest time. The values of $\overline{PTA}$ rise from <0.2 to 0.5 can be achieved by blocking 54%-68% solar irradiance among the four sites. However, when $\overline{PTA}$ <0.3, the improvement over baseline is difficult to achieve by enhancing the local wind speed by up to 10m/s, which is the maximum boost provided by typical ventilation fans. Enhancing the mechanical wind speed is effective to improve $\overline{PTA}$ at the time when $\overline{PTA}$ >0.3 (e.g., the cloudy day). Coupling the shading and boosting wind can be a more efficient way to improve $\overline{PTA}$. For example, increasing a 2 m/s wind speed requires a smaller shelter that blocks ~10% less solar irradiance as compared to a shading only solution to maintain $\overline{PTA}$ ≥0.5. This combination provides more flexibility for architecture based on site constraints. Moreover, the quantitative analysis of $\overline{PTA}$ and the well-defined uncomfortable region, purple zones as depicted in Figs 9(C), 10(C) and 12(C), can help develop strategies to improve environmental comfort to encourage outdoor gatherings and establish the reference of outdoor retail options for each site. With the aid of thermal and acoustic comfort evaluation results derived from IEM, specific outdoor retail space design recommendations for each site were suggested. The site (i) could be a naturally ventilated shopping street with awnings or overhangs extended out from the buildings to enhance the thermal comfort for visitor queuing. Retail spaces are not recommended for the site (ii), especially for the northern section with poor thermal and acoustic comfort. They could function as a pedestrian-only path with roadside trees or extensive shelters to improve the thermal comfort as a much higher amount of shading as compared to the site (i). The site (ii) is required to achieve the same amount of thermal acceptability from 50% of the visitors or pedestrians walking along this pedestrian thoroughfare. The site (iii) benefits from the shade natural noise barrier from three sides through the clever design of the building architecture. Thus, this makes this site an ideal location for long-term al fresco dining and/or as a podium for small-scale community events. The site (iv) exhibits thermal and acoustic comfort issues, and only short-term retail activities and community events with plenty of shading, such as the weekend bazaar held under tentages, are recommended. If long-term retail spaces are to be situated at this site, the

space must be air-conditioned to mitigate long-term thermal stress and noise impact on the staff working here. Our findings show that the design of the outdoor retail spaces for a tropical urban district is generally divided into three types of spaces: (1) outdoor spaces with no thermal and acoustic comfort issues (blue zone), (2) spaces with thermal comfort issues only (red zone), and (3) spaces with significant thermal and acoustic comfort issues (purple zone). The general retail options recommended for blue zones are al fresco dining or outdoor barbeque restaurants. For the red zones, naturally ventilated retail spaces, shopping streets with solar-blocking standalone shelters, awnings, or overhangs from buildings are preferred. With the expected high traffic noise level and thermally unacceptable environment in the purple zones, only short-term naturally ventilated retail spaces, e.g., weekend bazaars or short-term pop-up kiosks, should be considered. This will avoid extended periods of high-level noise exposure, even though the thermal comfort could be improved with shading infrastructure or active mechanical ventilation. However, if retail stores or restaurants must be included in the purple zones, these spaces (purple zone) must be enclosed and air-conditioned to keep out both traffic noise and ambient heat to minimize the health risk posed to the staff and regular patrons at these retail spaces with prolonged exposure. Furthermore, this study reveals that both thermal and acoustic comfort, derived from IEM solar-wind-noise coupled mapping, presents a previously under-explored dimension of retail space planning. Potential future works include validating IEM results against on-site measurements and evaluating possible landscape and infrastructure combinations that could help mitigate outdoor thermal and acoustic comfort issues locally (e.g., trees with shelter walkaways, green walls with outdoor ventilation fans, etc.).

## Supporting information

**S1 File.**
(ZIP)

## Author Contributions

**Conceptualization:** Po-Yen Lai, Wee Shing Koh.

**Data curation:** Po-Yen Lai, Dias Leong, Hyosoo Lee.

**Funding acquisition:** Wee Shing Koh.

**Investigation:** Po-Yen Lai, Wee Shing Koh, Harish Gopalan.

**Methodology:** Po-Yen Lai, Harish Gopalan, Huizhe Liu, Dias Leong, Hyosoo Lee, Gibert Peh, Gilbert Cher, Cheng Hui Eng, Jia Li Goh.

**Project administration:** Wee Shing Koh, Johnathan Goh.

**Resources:** Edmund Tan, James Tan.

**Software:** Harish Gopalan, Huizhe Liu, Dias Leong, Hyosoo Lee, Jacob Ang.

**Supervision:** Wee Shing Koh, Edmund Tan, James Tan.

**Validation:** Po-Yen Lai.

**Visualization:** Po-Yen Lai, Dias Leong, Hyosoo Lee, Jiun Yeu Lim.

**Writing – original draft:** Po-Yen Lai.

**Writing – review & editing:** Wee Shing Koh, Harish Gopalan, Huizhe Liu, Dias Leong, Hyo-soo Lee, Johnathan Goh, Jiun Yeu Lim, Jacob Ang, Gibert Peh, Gilbert Cher, Cheng Hui Eng, Jia Li Goh, Edmund Tan, James Tan.

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
