## [Decision Letter · Decision Letter 0]

18 Aug 2022

PONE-D-22-17882Outdoor environmental comfort evaluation for retail planning in a tropical business district using Integrated Environmental ModellerPLOS ONE

Dear Dr. Lai,

Thank you for submitting your manuscript to PLOS ONE. After careful consideration, we feel that it has merit but does not fully meet PLOS ONE’s publication criteria as it currently stands. Therefore, we invite you to submit a revised version of the manuscript that addresses the points raised during the review process.

We look forward to receiving your revised manuscript.

Kind regards,

Fausto Cavallaro, PhD

Academic Editor

PLOS ONE

2. Please note that PLOS journals require authors to make all data necessary to replicate their study’s findings publicly available without restriction at the time of publication. Please see our Data Availability policy at https://journals.plos.org/plosone/s/data-availability. As such, please make the full specific dataset used in this study available by either A) uploading the full dataset as supplementary information files, or B) including a URL link in your Data Availability Statement and Methods section to where the full dataset can be accessed.

Additionally, please note that PLOS ONE has specific guidelines on code sharing for submissions in which author-generated code underpins the findings in the manuscript. In these cases, all author-generated code must be made available without restrictions upon publication of the work. Please review our guidelines at https://journals.plos.org/plosone/s/materials-and-software-sharing#loc-sharing-code and ensure that your code is shared in a way that follows best practice and facilitates reproducibility and reuse.

    "This research is supported by A*STAR under its URBAN & GREENTECH OFFICE (UGTO) CONSULTANCY FUND (Grant No.: UGTO/CF/01). This development work was undertaken as a part of the project “Modeling of Air Flow, Thermal and Chemical Gas Dispersion Towards Next Generation Port (Tuas Maritime Hub)", funded by the Singapore Maritime Institute (SMI-2016-MA-04). The conclusions put forward reflect the views of the author alone, and not necessarily those of the funding institution"

 "This research is supported by A*STAR under its URBAN & GREENTECH OFFICE (UGTO) CONSULTANCY FUND (Grant No.: UGTO/CF/01). This development work was undertaken as a part of the project “Modeling of Air Flow, Thermal and Chemical Gas Dispersion Towards Next Generation Port (Tuas Maritime Hub)", funded by the Singapore Maritime Institute (SMI-2016-MA-04). The conclusions put forward reflect the views of the author alone, and not necessarily those of the funding institution. The funders had no role in study design, data collection and analysis, decision to publish, or preparation of the manuscript."

Reviewers' comments:

Reviewer's Responses to Questions

**Comments to the Author**

1. Is the manuscript technically sound, and do the data support the conclusions?

Reviewer #1: Yes

Reviewer #2: Partly

2. Has the statistical analysis been performed appropriately and rigorously? 

Reviewer #1: Yes

Reviewer #2: I Don't Know

3. Have the authors made all data underlying the findings in their manuscript fully available?

Reviewer #1: Yes

Reviewer #2: No

4. Is the manuscript presented in an intelligible fashion and written in standard English?

Reviewer #1: Yes

Reviewer #2: Yes

5. Review Comments to the Author

Reviewer #1: The flow of the paper is well organized and easy to follow. The introduction is well told and states all the research has been done. Research gap is clear connecting the objective of this paper. Some background on the research location however lacking on the details on the datasets used. Please provide more information on the dataset used. The results are very clear using the graph. The author can include studies from different paper as comparisons on with these paper findings if there is any. There are several typo in the writings, please recheck and send the paper for the proofreading.

Include all the keywords inside the Abstract. “Tropical urban district”, “Thermal comfort index”, “Acoustic comfort index”, “Environmental evaluation” and “retail planning” are not in the abstract.

Since coupled wind-thermal is one of main research, if possible the author might to explain more on definition or background what is coupled wind-thermal ?

Always start introduce the abbreviation first before use it. Line 64, “SOLWEIG” and “ENVI-met” not defined. Line 89, “WHO”. Line 104, “3D”.

Line 111, since author already defined IEM, just use IEM instead of full name.

The author claim there is no studies on the coupled solar-wind-noise simulations, however there is model developed and used to study for this case ? Seems like contradict. ( on line 67 – 71).

The reason of author choosing those specific research place ?

The author might consider to include yearly temperature to support author chosen day.

Table 3, how the author choose the direction ?

Line 287, “Fk” referring for three things ?

Line 318, reference on choosing db value ?

Line 366, the reason of choosing this specific initial temperature ?

Figure 5, no arrow in (b).

Line 442, reference defined the thresholds for thermal and acoustic comfort ? Same goes to threshold for PTA and PNA.

Line 612, is there any reference to support author claim on blocking solar.

Line 667, the author can include the reasoning using this ideal condition.

Maybe the author can compile results from Figure 9, 10 ,11 and 12 in Table.

Here are several reference the author might consider looking at:

• Field study of pedestrians’ comfort temperatures under outdoor and semi-outdoor conditions in Malaysian university campuses Othman, N.E., Zaki, S.A., Rijal, H.B., Ahmad, N.H., Razak, A.A. 2021. International Journal of Biometeorology. 65(4), pp. 453-477

• Analysis of urban morphological effect on the microclimate of the urban residential area of Kampung Baru in Kuala Lumpur using a geospatial approach. Open Access. Zaki, S.A., Azid, N.S., Shahidan, M.F., (...), Ali, M.S.M., Yakub, F. 2020. Sustainability (Switzerland). 12(18),7301

• Assessment of outdoor air temperature with different shaded area within an urban university campus in hot-humid climate. Open Access. Zaki, S.A., Syahidah, S.W., Shahidan, M.F., (...), Hassan, M.Z., Daud, M.Y.M. 2020. Sustainability (Switzerland). 12(14),5741, pp. 1-24

• Effects of urban morphology on microclimate parameters in an urban university campus. Open Access. Zaki, S.A., Othman, N.E., Syahidah, S.W., (...), Shahidan, M.F., Saudi, A.S.M. 2020. Sustainability (Switzerland). 12(7),2962

• Effects of roadside trees and road orientation on thermal environment in a tropical City. Open Access. Zaki, S.A., Toh, H.J., Yakub, F., (...), Ardila-Rey, J.A., Muhammad-Sukki, F. 2020. Sustainability (Switzerland). 12(3),1053

Reviewer #2: This study conducted a simulation-based assessment of outdoor thermal and acoustic comfort for a planned business urban district in Singapore. However, this study does not include the novelty and significance of their research. This study is close to a technical report rather than a research paper. In addition, the acoustic comfort evaluation was simply conducted.

6. PLOS authors have the option to publish the peer review history of their article (what does this mean?). If published, this will include your full peer review and any attached files.

Reviewer #1: No

Reviewer #2: No

---

## [Author Response · Author response to Decision Letter 0]

19 Oct 2022

Please see our bullet-point responses for all reviewer's comments in "Response to Reviewers".

---

## [Decision Letter · Decision Letter 1]

21 Dec 2022

PONE-D-22-17882R1Outdoor environmental comfort evaluation for retail planning in a tropical business district using Integrated Environmental ModellerPLOS ONE

Dear Dr. Lai,

Thank you for submitting your manuscript to PLOS ONE. After careful consideration, we feel that it has merit but does not fully meet PLOS ONE’s publication criteria as it currently stands. Therefore, we invite you to submit a revised version of the manuscript that addresses the points raised during the review process.

We look forward to receiving your revised manuscript.

Kind regards,

Fausto Cavallaro, PhD

Academic Editor

PLOS ONE

Journal Requirements:

Reviewers' comments:

Reviewer's Responses to Questions

**Comments to the Author**

1. If the authors have adequately addressed your comments raised in a previous round of review and you feel that this manuscript is now acceptable for publication, you may indicate that here to bypass the “Comments to the Author” section, enter your conflict of interest statement in the “Confidential to Editor” section, and submit your "Accept" recommendation.

Reviewer #1: All comments have been addressed

Reviewer #3: All comments have been addressed

2. Is the manuscript technically sound, and do the data support the conclusions?

Reviewer #1: Yes

Reviewer #3: Partly

3. Has the statistical analysis been performed appropriately and rigorously? 

Reviewer #1: Yes

Reviewer #3: I Don't Know

4. Have the authors made all data underlying the findings in their manuscript fully available?

Reviewer #1: Yes

Reviewer #3: Yes

5. Is the manuscript presented in an intelligible fashion and written in standard English?

Reviewer #1: Yes

Reviewer #3: No

6. Review Comments to the Author

Reviewer #1: The author answered all the questions and made the required changes. However, one of the suggestion paper is not included in this paper as follows:

Analysis of urban morphological effect on the microclimate of the urban residential area of Kampung Baru in

Kuala Lumpur using a geospatial approach. Open Access. Zaki, S.A., Azid, N.S., Shahidan, M.F., (...), Ali, M.S.M., Yakub,

F. 2020. Sustainability (Switzerland). 12(18),730

Reviewer #3: The authors are requested to address the following issues in the manuscript

1. clarify if the IEM simulation results are validated in the study

2. please correct the typographical and grammatical errors in the manuscript

3. Based on the study, certain generic recommendations can be provided in the discussion / conclusion.

4. In the Conclusion first few sentences are based on the methodology adopted which could be removed and it can begin with the specific recommendations based on the findings of the study and shall conclude with generic suggestion for future studies.

7. PLOS authors have the option to publish the peer review history of their article (what does this mean?). If published, this will include your full peer review and any attached files.

Reviewer #1: No

Reviewer #3: No

---

## [Author Response · Author response to Decision Letter 1]

12 Jan 2023

Please see the uploaded "Response to Reviewers" to find our responses.

---

## [Decision Letter · Decision Letter 2]

8 Feb 2023

Outdoor environmental comfort evaluation for retail planning in a tropical business district using Integrated Environmental Modeller

PONE-D-22-17882R2

Dear Dr. Lai,

We’re pleased to inform you that your manuscript has been judged scientifically suitable for publication and will be formally accepted for publication once it meets all outstanding technical requirements.

Kind regards,

Fausto Cavallaro, PhD

Academic Editor

PLOS ONE

Additional Editor Comments: The authors addressed all the reviewers comments. The paper now results improved.

Reviewers' comments:

Reviewer's Responses to Questions

**Comments to the Author**

1. If the authors have adequately addressed your comments raised in a previous round of review and you feel that this manuscript is now acceptable for publication, you may indicate that here to bypass the “Comments to the Author” section, enter your conflict of interest statement in the “Confidential to Editor” section, and submit your "Accept" recommendation.

Reviewer #1: All comments have been addressed

2. Is the manuscript technically sound, and do the data support the conclusions?

Reviewer #1: Yes

3. Has the statistical analysis been performed appropriately and rigorously? 

Reviewer #1: Yes

4. Have the authors made all data underlying the findings in their manuscript fully available?

Reviewer #1: Yes

5. Is the manuscript presented in an intelligible fashion and written in standard English?

Reviewer #1: Yes

6. Review Comments to the Author

Reviewer #1: The author already addressed all the comments accordingly. Therefore, this paper should be published accordingly.

7. PLOS authors have the option to publish the peer review history of their article (what does this mean?). If published, this will include your full peer review and any attached files.

Reviewer #1: No

---

## [Editor Report · Acceptance letter]

23 Feb 2023

PONE-D-22-17882R2 

Outdoor environmental comfort evaluation for retail planning in a tropical business district using Integrated Environmental Modeller 

Dear Dr. Lai:

I'm pleased to inform you that your manuscript has been deemed suitable for publication in PLOS ONE. Congratulations! Your manuscript is now with our production department. 

Kind regards, 

on behalf of

Professor Fausto Cavallaro 

Academic Editor

PLOS ONE